# The Association between Plant-Based Diet Indices and Metabolic Syndrome in Chinese Adults: Longitudinal Analyses from the China Health and Nutrition Survey

**DOI:** 10.3390/nu15061341

**Published:** 2023-03-09

**Authors:** Yating Huo, Suixia Cao, Jingchun Liu, Binyan Zhang, Kun Xu, Yutong Wang, Huimeng Liu, Peiying Yang, Lingxia Zeng, Hong Yan, Shaonong Dang, Baibing Mi

**Affiliations:** Department of Epidemiology and Biostatistics, School of Public Health, Xi’an Jiaotong University Health Science Center, No. 76, Yanta West Road, Xi’an 710061, China

**Keywords:** plant-based diets, metabolic syndrome, BMI, abdominal obesity

## Abstract

Objectives: To investigate the association between a plant-based diet and metabolic syndrome (MetS) among Chinese adults. Methods: Based on the data from the 2004–2015 China Health and Nutrition Survey and the corresponding edition of China Food Composition, we calculated the healthy plant-based diet indices (hPDI) and unhealthy plant-based diet indices (uPDI). The Cox proportional hazards regression model was used to estimate the hazard ratios (HRs) with 95% confidence intervals (CIs) for MetS. Mediation analysis was further conducted to explore the mediator role of Body Mass Index (BMI) in the association between hPDI and MetS. Results: We included 10,013 participants, and over a median follow-up of 5 years, 961 patients (9.60%) developed MetS. Compared to those in the lowest quintile of hPDI score, we found that those in the highest quintile of hPDI score had a 28% lower ([HR]: 0.72, 95% CI 0.56–0.93, *P*_trend_ = 0.021) risk of developing MetS and had a 20% lower (hazard ratio [HR]: 0.80, 95% CI 0.70–0.92, *P*_trend_ = 0.004) risk of developing abdominal obesity. No significant associations were observed between uPDI and the MetS, but those in the highest quintile of uPDI score had a 36% higher (hazard ratio [HR]: 1.36, 95% CI 1.20–1.64, *P*_trend_ < 0.001) risk of developing abdominal obesity, compared to those in the lowest quintile of uPDI score. In exploratory analysis, we observed that BMI at baseline mediated 27.8% of the association between hPDI and incident MetS, and BMI at baseline mediated 29.7% of the association between hPDI and abdominal obesity. Conclusion: The current findings reveal a possible causal relationship between a healthy plant-based diet and a reduced risk of MetS, especially abdominal obesity. It is observed that BMI may mediate the relationship between hPDI score and MetS. Controlling early dietary patterns and BMI may help reduce the risk of MetS.

## 1. Introduction

Metabolic syndrome (MetS) includes abdominal obesity, high blood glucose, hypertriglyceridemia, low high-density lipoprotein cholesterol (HDL-C), and elevated blood pressure [1,2]. It is a set of components that reflect excess nutrition and poor lifestyles, and the resulting obesity is associated with other comorbidities, including a prethrombotic state, pro-inflammatory state, metabolic-associated fatty liver disease, and reproductive disorders [3]. There is an increasing prevalence of MetS in the Asia-Pacific region, affecting more than 25% of adults in most countries [4]. A recent systematic review found that 24.5% of Chinese subjects aged 15 years and older were at risk of MetS [5]. This increase is associated with the global epidemic of obesity and diabetes [6]. Studies have shown that the accumulation of intra-abdominal adipose tissue is an essential determinant of MetS and that abdominal obesity is the most closely related to MetS [7]. In addition, reducing energy intake and increasing physical activity are the most basic ways to lose weight and reduce the risk of abdominal obesity [8].

A healthy diet plays an important role in preventing disease. It has been found that individuals following a vegan diet have a positive effect on weight loss [9], but the general population may not readily accept a strict vegan or vegetarian diet. It is well-established that preference for plant-derived foods but not the exclusion of animal foods has been associated with many diseases [10,11,12,13,14], such as diabetes, cardiovascular diseases, or MetS. This plant-based diet commonly varies from flexitarian to pescatarian or lacto/ovo-vegetarian [15]. Our study used the plant-based diet indices because its definition is quantitative and has been studied extensively [16,17]. Previous studies have found that healthy plant-based diet indices (hPDI) and unhealthy plant-based diet indices (uPDI) had different effects on health outcomes [18]. We hypothesized that there would be different effects on MetS.

Dietary and lifestyle changes have significant implications for the development of MetS [19]. Many observational studies have analyzed the association between diet and MetS [20,21,22]. However, there is still a lack of extensive prospective cohort studies to explore the association between a plant-based diet and MetS in China. Considering that the traditional Chinese diet is mainly based on plant foods, mostly vegetables and staple food [23], it is of great practical significance to explore the relationship between plant-based food and MetS in the Chinese population.

In this study, we evaluated the associations between two plant-based diet indices (hPDI and uPDI) with the risk of developing MetS in a nationally representative sample of Chinese adults.

## 2. Methods

### 2.1. Study Design and Study Population

The China Health and Nutrition Survey (CHNS) is a longitudinal survey that includes nutrition, economics, geographic location, and citizen health. With a multistage random clustering design, it was initiated in 1989 and followed up every 2 to 4 years. CHNS has completed ten rounds (1989, 1991, 1993, 1997, 2000, 2004, 2006, 2009, 2011, and 2015) [24]. In the 2009 wave of CHNS, blood samples were collected and analyzed in a national central lab in Beijing, China (medical laboratory accreditation certificate ISO 15189:2007) with strict quality control [25]. The institutional review boards of the University of North Carolina at Chapel Hill, the National Institute of Nutrition and Food Safety, and the Chinese Center for Disease Control and Prevention approved the study. Each participant provided written informed consent. A more detailed description of physiological data measurement methods has been provided in other articles [26].

The present study was based on five rounds of CHNS data from 2004 to 2015 (2004, 2006, 2009, 2011, and 2015). The data recorded until the 2015 survey were used to determine the survival status and times of the subjects we included. We included 13,345 participants surveyed in at least 2 rounds with dietary data. The first round has been termed the baseline. We excluded 2258 participants younger than 18 years of age, 60 participants with implausibly total energy intake outlier (<500 kcal/d or >5000 kcal/d), and 772 participants with myocardial infarction or stroke or cancer because the diagnosis of chronic diseases may prompt individuals to change their dietary behaviors. Lastly, we excluded 242 participants who had MetS during the first 2 years of follow-up. Our final analysis sample was 10,013. Figure 1 shows the flow chart of the subject-selection process.

### 2.2. Plant-Based Diet Indices

The dietary data were collected using 3 consecutive 24 h recalls at the individual level. The consumption of food at the household level was determined by the weighing method [27]. We calculated two plant-based diet indices (hPDI and uPDI). Based on the responses, hPDI and uPDI are consistent with the previously introduced calculation methods [11,28,29]. All subjects’ first diet data were used as the baseline. For food classification, we refer to the China Food Composition Tables [30] and previous studies by Bo Chen [26] et al., and categorize all foods into 16 food groups, which is more suitable for the dietary characteristics of Chinese people (Appendix A). The plant-based diet index was calculated based on the 16 food groups. These food groups were classified as healthy plant-based food (whole grain, fruit, vegetables, nuts, legumes, tea, and coffee), less healthy plant-based food (refined grains, potatoes, sugar-sweetened beverages, sweets and desserts, fermented food groups), and animal food (animal fat, dairy, eggs, fish, and meat).

To avoid the influence of food consumption, the residual method was used to adjust the total energy intake [31]. We ranked the participants in each food group by their food intake quintile. For the hPDI, positive scores were assigned to healthy plant-based food, and negative scores were assigned to less healthy plant-based food and animal food. For example, individuals in the highest quintile of fruit consumption scored five points, and those in the lowest quintile scored one point. For the uPDI, healthy plant-based food and animal food were negatively scored and less healthy plant-based food was positively scored. For example, individuals in the highest quintile of fruit consumption scored one point, and those in the lowest quintile scored five points. The scores of the 16 food groups were then summed for each participant to obtain the index, ranging from 16 to 90, with a higher index representing participants’ better adherence.

### 2.3. Outcome Assessment

We defined MetS according to the guidelines for preventing and controlling type 2 diabetes in China (2017 Edition) [32]. People with three or more of the following are diagnosed with MetS: (1) Abdominal obesity (central obesity): waist circumference ≥ 90 cm for men, ≥ 85 cm for women. (2) Hyperglycemia: fasting blood glucose ≥ 6.1 mmol/L or blood glucose ≥ 7.8 mmol/L 2 h after glucose load and/or diagnosed with diabetes and treated. (3) Hypertension: blood pressure ≥ 130/85 mmHg and/or hypertension confirmed and treated. (4) Fasting triglyceride (TG) ≥ 1.70 mmol/L. (5) Fasting high-density lipid cholesterol (HDL-C) < 1.04 mmol/L.

### 2.4. Covariates

Based on earlier knowledge and research [12], we included demographic characteristics (age, gender, and education) and lifestyle factors (smoking status, alcohol intake, physical activity, total energy intake (kcal), total carbohydrate intake (g), total fat intake (g), and total protein intake (g)). Education was divided into 0 to 6 years, 6 to 11 years, and ≥12 years. We calculated each participant’s metabolic equivalent of energy (MET) per day by accounting for types and intensity of physical activity [33]. The smoking status and alcohol intake were divided into yes or no, and Body Mass Index (BMI) was calculated as weight divided by the square of the height (kg/m^2^). Based on the recommended cut-off points for Chinese adults, BMI was divided into lean (<18.5 kg/m^2^), normal (18.5~23.9 kg/m^2^), overweight (24~27.9 kg/m^2^), and obesity (≥28 kg/m^2^) [34]. Models were adjusted for age, physical activity, total energy intake (kcal), total carbohydrate intake (g), total fat intake (g), total protein intake (g), and physical activity as continuous variables.

### 2.5. Statistical Analysis

Subjects were categorized into five groups according to the quintiles of the hPDI score and uPDI score. Continuous variables are presented as the mean ± standard deviation (SD); categorical variables are presented as percentages. We examined the differences using ANOVA for continuous variables and the Chi-squared test for categorical variables.

We used Cox proportional hazards models to evaluate the associations between plant-based diets and incident MetS. The length of follow-up time was used as the time metric. Firstly, it was a crude model. Then, in the adjusted model, we adjusted for age, gender, education, physical activity, smoking status, alcohol intake, total energy intake, total carbohydrate intake (g), total fat intake (g), and total protein intake (g). We analyzed the association of the quintiles of plant-based diets and MetS. Next, we examined whether plant-based diets were associated with individual components of MetS (abdominal obesity, high fasting glucose, hypertriglyceridemia, low HDL-C, and elevated blood pressure).

To test potential nonlinear associations, we used the lowest quintile of hPDI as a reference value to test the linear trend. Using the hPDI score as a continuous variable, we performed a restricted cubic spline (RCS) fit on the fully adjusted model to observe further the relationship between the healthy plant-based diet indices and MetS. Mediation analysis with multiple imputations, resamples, and adjustment for covariates was performed to evaluate whether BMI mediated the relationship between hPDI and MetS. Additionally, the “regiment” package in R software was used to estimate indirect, direct, and total effects [35]. The pure natural direct effect (PNDE) is typically called the natural direct effect (NDE). The total natural indirect effect (TNIE) is the corresponding natural indirect effect (NIE). The areas under the curve (AUC) of the receiver operating characteristic curve (ROC) analyses were used to evaluate the predictive performance of the hPDI and uPDI.

Subgroup analysis included categorical variables such as age, sex, and BMI to explore whether there were statistical differences between different groups and MetS. In the sensitivity analysis, we performed multiple imputations of years of education (5.08% missing), smoking status (5.03% missing), drinking status (5.03% missing), physical activity (25.08% missing), BMI (14.22% missing), total carbohydrate intake (5.03% missing), total fat intake (5.03% missing), and total protein intake (5.03% missing), using the FCS method [36]. Twenty imputed datasets were generated, and the final results were formed by combining the parameters of each imputed dataset computed using the standard statistical method.

All analyses were conducted with SAS 9.4 software (SAS Inc., Cary, NC, USA) and R version 4.2.1 software (R Foundation for Statistical Computing (https://www.r-project.org (accessed on 12 August 2021))) with “regiment” packages. *p* < 0.05 was considered statistically significant.

## 3. Results

### 3.1. Baseline Characteristics

The hPDI score ranged from 34 to 65, and the uPDI score ranged from 28 to 65 in all participants. Table 1 shows the baseline characteristics of the participants by quintiles of hPDI and uPDI scores. The higher quintile of hPDI was more likely to be male and relatively younger. Most of them lived in rural areas and were poorly educated, and fewer were obese. People in the highest quartile of hPDI tend to have a lower percentage of smokers and drinkers and have a high level of physical activity, and they have higher total energy and carbohydrate intake but lower fat intake. The higher quartile of uPDI was relatively younger and lived in rural areas. A higher proportion of them are overweight and smokers, and they had higher levels of physical activity and lower intake of total protein.

### 3.2. Association between Plant-Based Diet Indices and MetS

Over a median follow-up of 5 years, 961 patients (9.60%) developed MetS. Table 2 shows the hazard ratios (HRs) and 95% CIs of MetS according to the quintiles of hPDI and uPDI scores. In the crude model, those in the highest quintile of hPDI had a 37% lower ([HR]: 0.63, 95% CI 0.51–0.77, *p*-trend < 0.001) risk of developing MetS compared to those in the lowest quintile of hPDI. In the adjusted model, we adjusted for demographic characteristics, lifestyle factors, and dietary intake variables. This association was slightly attenuated but remained significant. Those in the highest quintile of hPDI had a 28% lower ([HR]: 0.72, 95% CI 0.56–0.93, *p*-trend = 0.021) risk of developing MetS than those in the lowest quintile of hPDI. However, no statistically significant differences were found between uPDI with MetS.

We also analyzed the relationship between hPDI and individual signs of MetS (abdominal obesity, high fasting glucose, hypertriglyceridemia, low HDL-C, and elevated blood pressure) in Table 2. In the adjusted model, we found that higher hPDI was associated with lower abdominal obesity. Those in the highest quintile of hPDI had a 20% lower (hazard ratio [HR]: 0.80, 95% CI 0.70–0.92, *p*-trend = 0.004) risk of developing abdominal obesity compared to those in the lowest quintile of hPDI. This association is reflected in our intuitive description of the relationship between hPDI and the incidence of MetS (Figure 2). Those in the highest quintile of uPDI had a 36% higher (hazard ratio [HR]: 1.36, 95% CI 1.20–1.54, *p*-trend < 0.001) risk of developing incident abdominal obesity, compared to those in the lowest quintile of uPDI.

### 3.3. Explore Analysis

After adjusting covariates, we found that BMI greatly impacted the outcome, and abdominal obesity was the main component affecting the outcome, so we further analyzed the mediating role of BMI on abdominal obesity and MetS. The result of mediation analysis is shown in Table 3. There were significant direct and indirect effects between hPDI and incident risk of MetS [PNDE = −0.019 (−0.022, −0.016), *p* < 0.0001 and TNIE = −0.007 (−0.010, −0.005), *p* < 0.0001], and BMI mediated 27.8% of the above association. Meanwhile, we observed that BMI mediated 29.7% of the association between hPDI and abdominal obesity [PNDE = −0.016 (−0.017, −0.014), TNIE = −0.007 (−0.008, −0.005)]). The full adjustment MetS prediction model yielded an AUC of 0.652 (95% CI 0.633–0.672) for hPDI and 0.654 (95% CI 0.635–0.673) for uPDI (Appendix A).

### 3.4. Subgroups and Sensitivity Analysis

As shown in the forest plot (Figure 3), in adults younger than 40 years of age, hPDI scores had a protective effect on MetS. In the female population, the protective effect was statistically significant. However, no statistical significance was found in the male population. Among non-drinkers, higher hPDI scores were associated with a statistically significant reduction in the risk of MetS. The protective effect was statistically significant for people with an energy intake of more than 2182.87 kcal per day or less than 11 years of education. Multiple imputations were used for sensitivity analysis (Appendix A), and the results were consistent with the previous results and remained robust, supporting that the increase in the hPDI score is a protective factor of MetS.

## 4. Discussion

In the CHNS cohort, we observed that the higher adherence to a healthy plant-based diet, the lower the risk of MetS, particularly abdominal obesity. The results remained significant after adjusting for other covariates, but the results of the unhealthy plant-based diet were not statistically significant. We found that BMI may mediate 27.8% of the association between hPDI and MetS.

Compared with the results of a study on a plant-based diet and MetS in South Korea [12], in the Korean Genome and Epidemiology Study (KoGES) cohort, when adjusting for demographic traits and lifestyle factors, those in the highest quintile of uPDI had a 44% greater risk of having incident MetS compared to those in the lowest quintile of uPDI, but they did not find an association between hPDI and MetS. Our study can supplement this research gap. In the “Seguimiento Universidad de Navarra” (SUN) prospective cohort study [16], they found a stronger inverse association between the healthful pro-vegetarian food pattern and overweight/obesity incidence, whereas these were not statistically significant for the unhealthy pro-vegetarian food pattern. The vegetarian food pattern was calculated similarly to the plant-based diet index in our study. Our study may further support this conclusion in a published study using the same data source from CHNS [26]. They analyzed the relationship between the plant-based diet index and obesity, hypertension, and type 2 diabetes. In contrast, the highlight of our study lies in the use of the blood sample data collected in 2009, in which fasting blood glucose, triglyceride, and high-density lipid cholesterol indexes are used, which makes up for the lack of prospective cohort studies on the causal relationship of MetS in China.

We did not find an association between uPDI and MetS, but previous studies have found an association between unhealthy plant-based foods and type 2 diabetes, cardiovascular disease, and MetS [11,12,29]. The majority of research on plant-based diets has been completed on people in western countries, where cooking methods mainly consist of pan frying, sautéing, and baking. In contrast, steaming, boiling, and stir-frying are more common in Chinese cooking methods [37]. Studies have shown that cooking methods have a strong influence on carotenoid composition and bioavailability of vegetables [38]. In addition, many western diets tend to include larger portion sizes, which can lead to excessive energy intake and an increased risk of obesity [39]. Moreover, Western dietary patterns characterized by high consumption of red meat, processed meat, and butter are associated with higher prevalence and incidence of MetS [40]. Chinese people eat much more plant-based foods than those in western countries, but fewer animal-based foods, so the results may be different. Compared to the South Korean diet, where fermented foods are an essential component of the Korean diet [41], Chinese people consume relatively little preserved food. It has been found that fermented plant food can increase the content of vitamins, minerals and phenolic compounds in food [42], but may also lead to excessive sodium intake. This may result in an insignificant relationship between uPDI score and MetS, and the ROC curves of hPDI and uPDI are similar.

In addition, we found gender differences in the plant-based diet index, similar to those found in previous studies. It was found that there was a stronger association between uPDI and MetS in women (OR: 1.62, 95% CI 1.26–2.09, *p*-trend = 0.01) than in men (OR: 1.35, 95% CI 1.03–1.76, *p*-trend = 0.02) (*p* for interaction = 0.20) [43]. Our findings found that hPDI had a significant protective effect on women, but no significant results were found in men. Another study from Taiwan found that a vegetarian diet was associated with a reduced risk of symptomatic gallstones in women compared with a non-vegetarian diet (hazard ratio [HR]: 0.52; 95% CI 0.28–0.96), but not for men [44]. Presumably, the disparities in the plant-based diet index between the sexes are likely caused by the different dietary habits of men and women. Meat is often associated with gendered descriptions of masculinity in traditional Chinese concepts [45], which may explain why vegetarianism is more prevalent among women. Our study did not further explore women’s preference for plant-based foods. However, a survey based on CHNS data showed that only 34.3% of Chinese women followed animal-based diets [46]. In addition, a cross-sectional study from the NutriNet-Santé study found that vegetarians, who were more likely to be female (85.0%), had a balanced diet of macronutrients, and a lower incidence of chronic disease than meat eaters [47], which may explain the observed protective effect of hPDI scores on MetS in women in our study.

In terms of macronutrients, plant-based diets are characterized by a high intake of carbohydrates, dietary fiber, and plant-based proteins, a low intake of saturated and trans fats, and a high intake of unsaturated fatty acids [15]. Plant-based diets have a low energy density and high fiber content [22]. A study of Spanish adults demonstrated that dietary fiber intake was associated with a reduction in body weight and abdominal obesity [48], and dietary fiber can significantly reduce energy intake [49,50]. Low-energy food intake is significant for weight control and can reduce the incidence of abdominal obesity [51,52,53]. Abdominal obesity has been strongly linked to diabetes, hypertension, cardiovascular diseases, cancer, kidney diseases, and non-alcoholic fatty liver diseases (NAFLDs) [54]. Our study found that healthy plant-based foods may reduce the risk of MetS primarily by reducing the risk of abdominal obesity. Our findings supported the idea that healthy plant-based foods such as whole grains, vegetables, and fruits may reduce the risk of developing abdominal obesity.

Different dietary patterns can affect the BMI of individuals [55,56]. Studies have found that Hispanic/Latino SDAs who ate a plant-based diet had a lower BMI than non-vegetarians [57]. A prospective cohort study by the National Health Insurance Service in Korea found U-shaped associations between baseline BMI and mortality from any cause, cardiovascular disease, and cancer after adjusting other covariates [58]. As the most widely used index in clinical practice, BMI has been shown to replace waist circumference as an indicator to predict MetS [59], indicating a strong correlation between BMI and MetS. However, the mediating role of baseline BMI between a plant-based diet and MetS is still lacking. In the CHNS cohort, we observed that BMI might mediate 27.8% of the association between hPDI and MetS, which supported this correlation.

The advantage of our study is that it uses a large sample of national data, so the data are representative, and the results are reliable. In addition, to our knowledge, it was the first study to investigate the associations between a plant-based diet and MetS in China. Several limitations merit mentioning when interpreting the results. First, since blood samples data were measured in 2009 and dietary data in this study started in 2004, the lack of blood samples data from some years may lead to the inability to draw significant conclusions, primarily affecting the association between the plant-based dietary index and triglycerides or high-density lipid cholesterol. This may also lead to the fact that the RCS curve between hPDI score and hypertriglyceridemia and low HDL-C was not statistically significant. Second, although we adjusted for multiple covariates, we could not control for some untested confounders and residual covariates, which could lead to confounding bias. Third, to avoid the effect of reverse causation in the cohort study, we excluded participants who had MetS during the first two years of follow-up but still could not completely avoid the effect of reverse causation. Finally, different cooking styles can have an impact on the nutritional value of plant-based foods [39], while the effects of cooking methods on the nutritional value of foods were not considered in the study.

## 5. Conclusions

Based on the Chinese population’s dietary characteristics, a healthy plant-based diet may reduce the risk of developing MetS, especially abdominal obesity. We found that BMI may mediate, so controlling early dietary patterns and BMI may be necessary to prevent MetS. Our study provides support for dietary patterns in the prevention of MetS.

## Figures and Tables

**Figure 1 nutrients-15-01341-f001:**
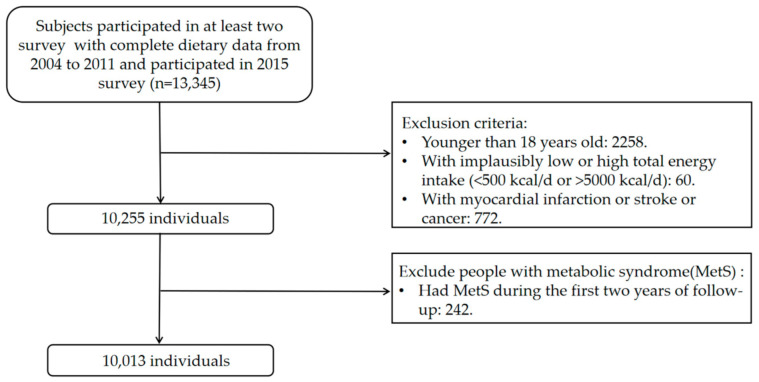
Flow chart of the subject-selection process.

**Figure 2 nutrients-15-01341-f002:**
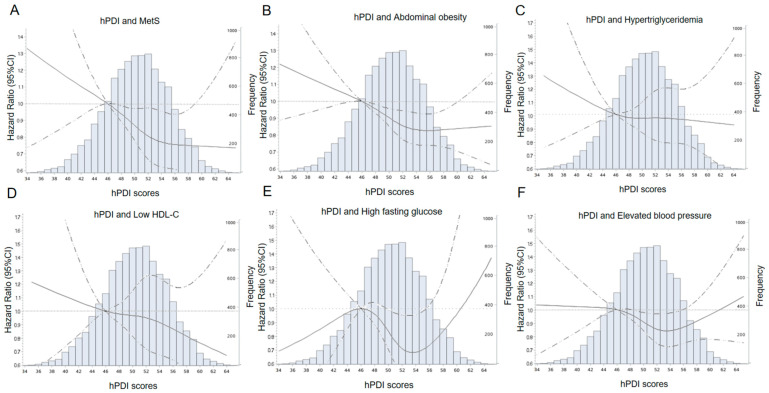
Adjusted hazard ratios and 95% confidence intervals for incident MetS (**A**), abdominal obesity (**B**), hypertriglyceridemia (**C**), low HDL-C (**D**), high fasting glucose (**E**), and elevated blood pressure (**F**) according to the continuous hPDI. Curves were fitted as a smooth term using a restricted cubic spline with 4 knots (5%, 35%, 65%, 95%). The solid lines represent the adjusted hazard ratios (HRs) for incident MetS and the signs of it. The dashed lines represent 95% confidence intervals. The reference value was the lowest quintile of hPDI. The histogram shows the distribution of hPDI. HRs were adjusted for age, sex, total energy intake (kcal/d), total carbohydrate intake (g), total fat intake (g), total protein intake (g), education, physical activity, smoking status, and alcohol intake. hPDI, healthful plant-based diet index; MetS, metabolic syndrome; HDL-C, high-density lipoprotein cholesterol.

**Figure 3 nutrients-15-01341-f003:**
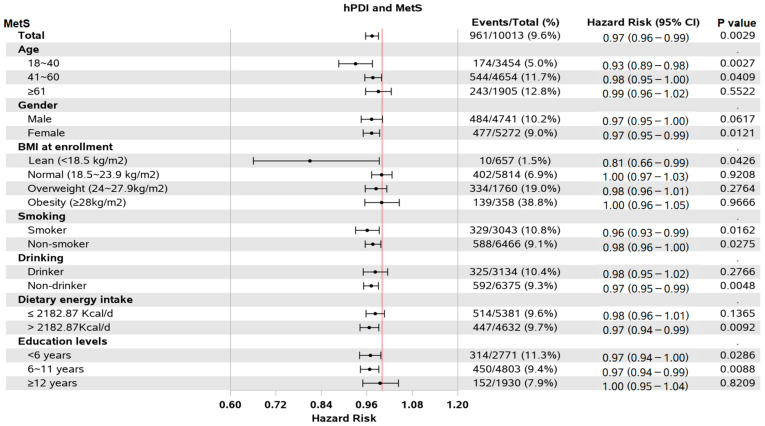
Forest plot of subgroups. The red line means that hazard ratio is equal to 1.hPDI, healthful plant-based diet index; MetS, metabolic syndrome.

**Table 1 nutrients-15-01341-t001:** Baseline characteristics of CHNS according to quintiles of plant-based diet indices (*n* = 10,013) ^a^.

Characteristics	Quintile 1	Quintile 2	Quintile 3	Quintile 4	Quintile 5	*p* Value ^b^
hPDI						
Sample size, n	2033	2175	2481	1366	1958	
Median score (range)	44 (34–46)	48 (47–49)	51 (50–52)	53 (53–54)	56 (55–65)	<0.001
Age, year	47.5 ± 15.0	46.3 ± 14.6	46.1 ± 14.7	46.9 ± 14.9	47.8 ± 14.4	<0.001
Female, %	1017 (50.0)	1059 (48.7)	1160 (46.8)	616 (45.1)	889 (45.4)	0.009
Urban, %	1006 (51.4)	786 (38.1)	767 (32.8)	335 (26.1)	406 (21.8)	<0.001
Education, year						
0–6	399 (20.4)	519 (25.1)	645 (27.6)	421 (32.8)	787 (42.2)	<0.001
6–11	972 (49.7)	1080 (52.3)	1216 (52.0)	665 (51.9)	870 (46.6)	
≥12	583 (29.8)	466 (22.6)	476 (20.4)	196 (15.3)	209 (11.2)	
BMI, kg/m^2^						
Lean (<18.5)	118 (6.7)	135 (7.2)	166 (7.9)	98 (8.5)	140 (8.2)	<0.001
Normal (18.5~23.9)	1104 (62.6)	1227 (65.8)	1444 (68.8)	799 (68.9)	1240 (72.8)	
Overweight (24~27.9)	442 (25.1)	411 (22.1)	395 (18.8)	226 (19.5)	286 (16.8)	
Obesity (≥28)	100 (5.7)	91 (4.9)	94 (4.5)	36 (3.1)	37 (2.2)	
Smoking, %	667 (34.1)	686 (33.2)	714 (30.5)	402 (31.4)	574 (30.7)	0.054
Drinking, %	738 (37.7)	678 (32.8)	755 (32.3)	378 (29.5)	585 (31.3)	<0.001
SBP, mmHg	121.1 ± 16.5	121.7 ± 17.4	120.4 ± 17.2	120.7 ± 18.3	120.6 ± 18.1	0.111
DBP, mmHg	78.9 ± 10.7	78.9 ± 10.8	77.9 ± 10.6	77.9 ± 10.7	77.8 ± 11.0	<0.001
Physical activity, MET-H/d	20.9 ± 16.4	22.1 ± 18.6	22.6 ± 18.0	22.7 ± 18.8	24.2 ± 19.6	<0.001
Healthy plant foods ^c^	9.6 ± 2.3	10.7 ± 2.1	11.7 ± 2.0	12.8 ± 1.8	14.2 ± 1.9	<0.001
Less-healthy plant foods ^d^	20.4 ± 2.3	21.7 ± 2.1	22.5 ± 1.8	23.1 ± 1.7	23.9 ± 1.4	<0.001
Animal foods ^e^	13.7 ± 3.0	15.6 ± 2.6	16.8 ± 2.2	17.6 ± 1.9	18.7 ± 1.6	<0.001
Total Energy, kcal	2177.7 ± 658.6	2149.3 ± 671.9	2142.2 ± 645.5	2188.1 ± 648.2	2273.4 ± 638.1	<0.001
Total Carbohydrate, g	277.6 ± 94.8	293.0 ± 98.3	311.7 ± 101.8	336.9 ± 109.7	371.2 ± 114.8	<0.001
Total Fat, g	83.2 ± 39.5	76.6 ± 39.5	68.2 ± 34.4	63.8 ± 33.4	55.8 ± 31.3	<0.001
Total Protein, g	71.9 ± 24.8	67.3 ± 24.1	64.2 ± 23.7	64.1 ± 22.5	65.7 ± 22.9	<0.001
uPDI						
Sample size, n	2674	1607	2416	1461	1855	
Median score (range)	43 (28–45)	47 (46–47)	49 (48–50)	51 (51–52)	55 (53–65)	<0.001
Age, year	47.0 ± 15.0	46.5 ± 14.5	46.7 ± 14.9	46.3 ± 14.4	45.2 ± 14.0	0.001
Female, %	1470 (55.0)	847 (52.7)	1299 (53.8)	737 (50.4)	919 (49.5)	<0.001
Urban, %	1139 (44.6)	568 (36.8)	764 (33.2)	383 (27.8)	446 (25.8)	<0.001
Education, year						
0–6	561 (22.0)	447 (28.9)	716 (31.1)	452 (32.8)	595 (34.5)	
6–11	1264 (49.6)	771 (49.9)	1171 (50.9)	702 (50.9)	895 (51.8)	<0.001
≥12	725 (28.4)	327 (21.2)	416 (18.1)	225 (16.3)	237 (13.7)	
BMI, kg/m^2^						
Lean (<18.5)	183 (7.9)	108 (7.8)	172 (8.3)	88 (7.0)	106 (6.8)	
Normal (18.5~23.9)	1555 (67.4)	962 (69.1)	1414 (67.8)	845 (67.6)	1038 (66.8)	0.577
Overweight (24~27.9)	470 (20.4)	259 (18.6)	418 (20.0)	266 (21.3)	347 (22.3)	
Obesity (≥28)	98 (4.3)	64 (4.6)	82 (3.9)	51 (4.1)	63 (4.1)	
Smoking, %	734 (28.8)	498 (32.2)	755 (32.8)	462 (33.5)	594 (34.4)	<0.001
Drinking, %	806 (31.6)	535 (34.6)	764 (33.2)	460 (33.3)	569 (33.0)	0.366
SBP, mmHg	120.6 ± 17.5	120.8 ± 16.9	121.0 ± 17.7	121.7 ± 18.1	120.8 ± 17.0	0.454
DBP, mmHg	77.7 ± 9.9	78.4 ± 10.2	78.2 ± 11.1	79.3 ± 10.7	79.1 ± 11.5	<0.001
Physical activity, MET-H/d	20.7 ± 15.9	21.6 ± 17.5	23.1 ± 18.8	23.8 ± 19.2	24.2 ± 20.4	<0.001
Healthy plant foods	22.5 ± 2.3	23.6 ± 2.1	24.4 ± 2.1	25.3 ± 2.0	26.6 ± 1.9	<0.001
Less-healthy plant foods	6.3 ± 1.6	6.8 ± 1.8	7.6 ± 1.8	8.5 ± 1.9	10.1 ± 2.1	<0.001
Animal foods	13.9 ± 2.8	16.2 ± 2.4	16.9 ± 2.3	17.7 ± 2.1	18.5 ± 1.7	<0.001
Total Energy, kcal	2191.2 ± 626.4	2183.1 ± 615.1	2167.5 ± 656.5	2198.1 ± 673.7	2178.6 ± 707.6	0.623
Total Carbohydrate, g	299.9 ± 100.0	316.6 ± 104.4	314.9 ± 107.9	331.1 ± 119.0	327.0 ± 113.2	<0.001
Total Fat, g	74.5 ± 35.4	69.0 ± 34.7	69.1 ± 37.7	67.3 ± 37.3	68.1 ± 40.6	<0.001
Total Protein, g	75.6 ± 23.95	67.1 ± 22.1	64.8 ± 23.4	62.6 ± 22.3	59.2 ± 23.1	<0.001

^a^ Data are means ± SDs for continuous variables and % for categorical variables. ^b^ Based on the ANOVA for continuous data and χ^2^ tests for categorical data. ^c^ Healthy plant foods are aggregated scores of whole grains, fruits, vegetables, nuts, legumes, tea, and coffee. ^d^ Less-healthy plant foods are aggregated scores of refined grains, potatoes, sugar-sweetened beverages, sweets and desserts, and fermented food groups. ^e^ Animal foods are aggregated scores of animal fat, dairy, eggs, fish, and meat. hPDI, health plant-based diet index; uPDI, unhealthful plant-based diet index. BMI, body mass index; SBP, systolic blood pressure; DBP, diastolic blood pressure; MET, metabolic equivalent of energy.

**Table 2 nutrients-15-01341-t002:** Hazard ratios (95% CI) of plant-based diet indices and incident MetS.

	Quintile 1	Quintile 2	Quintile 3	Quintile 4	Quintile 5	*p*-Trend
hPDI						
Incidence Mets						
cases/total	223/2016	232/2164	219/2472	115/1357	172/1940	
Crude model	Reference	0.91 (0.75, 1.09)	0.71 (0.59, 0.86)	0.64 (0.51, 0.81)	0.63 (0.51, 0.77)	<0.001
Adjusted model *	Reference	0.94 (0.76, 1.16)	0.74 (0.59, 0.93)	0.74 (0.57, 0.97)	0.72 (0.56, 0.93)	0.021
Abdominal obesity						
cases/total	723/2016	742/2164	807/2472	412/1357	616/1940	
Crude model	Reference	0.90 (0.81, 1.00)	0.79 (0.72, 0.87)	0.69 (0.62, 0.78)	0.68 (0.61, 0.76)	<0.001
Adjusted model	Reference	0.93 (0.83, 1.05)	0.88 (0.78, 0.99)	0.79 (0.68, 0.91)	0.80 (0.70, 0.92)	0.004
Hypertriglyceridemia						
cases/total	231/384	239/437	227/406	133/234	190/362	
Crude model	Reference	0.89 (0.74, 1.07)	0.89 (0.74, 1.07)	0.92 (0.75, 1.14)	0.82 (0.67, 0.99)	0.354
Adjusted model	Reference	0.91 (0.73, 1.13)	0.91 (0.72, 1.14)	0.95 (0.72, 1.23)	0.87 (0.68, 1.13)	0.844
Low HDL-C						
cases/total	130/384	116/437	122/406	63/234	90/362	
Crude model	Reference	0.78 (0.61, 1.00)	0.86 (0.67, 1.10)	0.79 (0.59, 1.07)	0.71 (0.54, 0.93)	0.110
Adjusted model	Reference	0.75 (0.55, 1.03)	0.92 (0.67, 1.27)	0.89 (0.61, 1.30)	0.74 (0.51, 1.08)	0.331
High fasting glucose						
cases/total	78/2016	96/2164	88/2472	39/1357	58/1940	
Crude model	Reference	1.04 (0.77, 1.41)	0.79 (0.59, 1.08)	0.59 (0.40, 0.87)	0.57 (0.41, 0.80)	<0.001
Adjusted model	Reference	1.21 (0.86, 1.70)	0.83 (0.57, 1.21)	0.89 (0.57, 1.38)	0.84 (0.55, 1.28)	0.224
Elevated blood pressure						
cases/total	796/2016	904/2164	973/2472	520/1357	863/1940	
Crude model	Reference	0.97 (0.89, 1.07)	0.85 (0.78, 0.94)	0.78 (0.69, 0.87)	0.84 (0.76, 0.92)	<0.001
Adjusted model	Reference	0.96 (0.86, 1.08)	0.91 (0.81, 1.02)	0.83 (0.73, 0.95)	0.94 (0.83, 1.06)	0.070
uPDI						
Incidence Mets						
cases/total	251/2662	137/1600	223/2399	154/1452	196/1836	
Crude model	Reference	0.88 (0.72, 1.09)	0.95 (0.79, 1.13)	1.04 (0.85, 1.28)	1.08 (0.89, 1.30)	0.398
Adjusted model	Reference	0.97 (0.76, 1.24)	1.07 (0.86, 1.33)	1.26 (0.99, 1.62)	1.31 (1.03, 1.66)	0.075
Abdominal obesity						
cases/total	821/2662	515/1600	777/2399	513/1452	674/1836	
Crude model	Reference	1.01 (0.90, 1.13)	1.02 (0.92, 1.12)	1.09 (0.98, 1.22)	1.10 (0.99, 1.22)	0.246
Adjusted model	Reference	1.16 (1.02, 1.31)	1.14 (1.02, 1.28)	1.35 (1.19, 1.54)	1.36 (1.20, 1.54)	<0.001
Hypertriglyceridemia						
cases/total	270/486	158/276	241/440	158/283	193/338	
Crude model	Reference	1.07 (0.88, 1.30)	0.97 (0.81, 1.15)	0.95 (0.78, 1.15)	1.05 (0.87, 1.26)	0.781
Adjusted model	Reference	1.05 (0.82, 1.32)	0.92 (0.74, 1.14)	0.93 (0.73, 1.19)	1.12 (0.88, 1.42)	0.465
Low HDL-C						
cases/total	130/486	78/276	112/440	95/283	106/338	
Crude model	Reference	1.07 (0.80, 1.41)	0.94 (0.73, 1.21)	1.18 (0.90, 1.54)	1.18 (0.91, 1.52)	0.361
Adjusted model	Reference	1.09 (0.76, 1.55)	0.97 (0.70, 1.33)	1.24 (0.88, 1.76)	1.37 (0.97, 1.94)	0.255
High fasting glucose						
cases/total	107/2662	54/1600	88/2399	52/1452	58/1836	
Crude model	Reference	0.83 (0.60, 1.15)	0.88 (0.67, 1.17)	0.85 (0.61, 1.18)	0.76 (0.55, 1.05)	0.521
Adjusted model	Reference	0.97 (0.67, 1.42)	1.08 (0.77, 1.50)	0.95 (0.63, 1.44)	0.91 (0.61, 1.38)	0.938
Elevated blood pressure						
cases/total	1053/2662	634/1600	964/2399	601/1452	804/1836	
Crude model	Reference	0.98 (0.89, 1.08)	0.99 (0.91, 1.08)	1.01 (0.91, 1.11)	1.04 (0.95, 1.14)	0.802
Adjusted model	Reference	1.05 (0.94, 1.18)	1.08 (0.97, 1.20)	1.16 (1.03, 1.31)	1.15 (1.02, 1.29)	0.092

* Adjusted model was adjusted for age, sex, total energy intake (kcal/d), total carbohydrate intake (g), total fat intake (g), total protein intake (g), education, physical activity, smoking status, and alcohol intake. hPDI, health plant-based diet index; uPDI, unhealthful plant-based diet index; MetS, metabolic syndrome; HDL-C, high-density lipoprotein cholesterol.

**Table 3 nutrients-15-01341-t003:** Mediation analysis of BMI.

Effect *	CDE (Controlled Direct Effect)	PNDE (Pure Natural Direct Effect)	TNIE (Total Natural Indirect Effect)	TNDE (Total Natural Direct Effect)	PNIE (Pure Natural Indirect Effect)	TE (Total Effect)	PM (Proportion Mediated)
BMI and MetS	−0.063 (−0.084, −0.041)	−0.019 (−0.022, −0.016)	−0.007 (−0.010, −0.015)	−0.019 (−0.022, −0.016)	−0.007 (−0.010, −0.005)	−0.026 (−0.030, −0.022)	0.278 (0.203, 0.353)
BMI and Abdominal obesity	−0.053 (−0.066, −0.042)	−0.016 (−0.017, −0.014)	−0.007 (−0.008, −0.005)	−0.016 (−0.018, −0.014)	−0.007 (−0.008, −0.005)	−0.022 (−0.025, −0.020)	0.297 (0.250, 0.343)

* Effect was adjusted for age, sex, total energy intake (kcal/d), total carbohydrate intake (g), total fat intake (g), total protein intake (g), education, physical activity, smoking status, and alcohol intake. Abbreviations: BMI, body mass index, MetS, metabolic syndrome.

## Data Availability

The data were obtained from CHNS. The original database is available at the website (https://www.cpc.unc.edu/projects/china), (accessed on 1 February 2023).

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
