# Peer review of "The Association between Plant-Based Diet Indices and Metabolic Syndrome in Chinese Adults: Longitudinal Analyses from the China Health and Nutrition Survey"

_nutrients, 2023, doi:10.3390/nu15061341_

Round 1

Reviewer 1 Report

In the manuscript submitted to me for review entitled The Association between Plant-Based Diet Indices and Metabolic Syndrome in Chinese Adults:A Longitudinal Analyses from the China Health and Nutrition Survey the authors examined the association between a plant-based diet and the development of the metabolic syndrome (MetS) among Chinese adults. The study was conducted over 11 years (2004–2015), involving 10,013 participants with an average follow-up period of 5 years for each participant.

My remarks and recommendations to the authors are:

1.     My suggestion is that in Figure 2, the individual figures within it should be labeled as A, B, C, etc., with each of the six figures indicating below the figure what information it represents. I think that this way the figure will be more accessible and easy to compare the presented results.

2.     Figure 3 needs to be enlarged and/or bolded to see the information it presents, because in this version the text inside it is too small and illegible.

3.     In the References, some of the literary sources do not list all the authors. Let the missing authors be filled. The ones I noticed were #16, #23, and #42.

Reviewer 2 Report

Comments and Suggestions for Authors

The manuscript is an original article that assesses the association between plant-based diet indices and metabolic syndrome in Chinese adults. The topic is of great interest to clinicians, nutritionists, and basic researchers, and the manuscript is well-written in terms of the English language. The paper provides a generous introduction and sufficient background to understand its message. Although similar studies were conducted in Europe, the USA, and Korea, this study stands out for its longitudinal design over 5 years and a large number of participants that validates its findings for the Chinese population. 

There are only a few issues that should be addressed:

  1. Lines 47-48: the phrase is unclear. Please specify exactly why it is beneficial “….to reduce energy intake, lose weight and increase physical activity”.
  2. Line 97: the sentence “Based on the sixth edition of the Chinese Food Composition Table.” is unclear (probably the verb is missing).
  3. There is an interesting finding (lines 206-207) …” no statistically significant differences were found between uPDI with MetS.” which the authors tried to explain in the discussions (lines 283-297) by the difference between the European diet (based on bread and cereals as a source of fiber) and the Chinese diet (based on plant food and less animal food). Instead, the references given (11, 12, 29) are studies from USA and Korea that found an association between unhealthy plant-based foods and MetS. In other words, either insert the appropriate references or try to give another explanation.
  4. Line 208: the term “composition” (of MetS) is inappropriate, because abdominal obesity, high fasting glucose, hypertriglyceridemia, low HDL-C, and elevated blood pressure are all signs of MetS. Please change it accordingly.
  5. Line 251: “However, no statistical significance was found in the male” regarding the protective effect of hPDI scores on MetS.  This is another interesting finding that the authors attempt to explain in the discussion section (lines 306-307), but without issuing any hypothesis in relation to it.
  6.  Please add “the effects of cooking methods on the nutritional value of foods were not considered in the study” to the limitations of the study.

Overall, it was a pleasure to read this well-structured and clear article.
